# Matrix Metalloproteinase-1 (MMP1) Upregulation through Promoter Hypomethylation Enhances Tamoxifen Resistance in Breast Cancer

**DOI:** 10.3390/cancers14051232

**Published:** 2022-02-27

**Authors:** Hyeon Woo Kim, Jae Eun Park, Minjae Baek, Heejoo Kim, Hwee Won Ji, Sung Hwan Yun, Dawoon Jeong, Juyeon Ham, Sungbin Park, Xinpei Lu, Han-Sung Kang, Sun Jung Kim

**Affiliations:** 1Department of Life Science, Dongguk University-Seoul, Goyang 10326, Korea; opopr5@dongguk.edu (H.W.K.); 201511717@dongguk.edu (J.E.P.); minjae1644@dgu.ac.kr (M.B.); hjkim@olixpharma.com (H.K.); hweewon96@naver.com (H.W.J.); sunghwan151019@gmail.com (S.H.Y.); hn999531@gmail.com (D.J.); wndus1000@kogene.co.kr (J.H.); do31100@gmail.com (S.P.); 2State Key Laboratory of Advanced Electromagnetic Engineering and Technology, School of Electrical and Electronic Engineering, Huazhong University of Science and Technology, Wuhan 430074, China; luxinpei@hust.edu.cn; 3Research Institute and Hospital, National Cancer Center, Goyang 10408, Korea; rorerr@ncc.re.kr

**Keywords:** breast cancer, MMP1, CpG methylation, tamoxifen resistance, xenograft

## Abstract

**Simple Summary:**

Cancer recurrence caused by tamoxifen resistance hampers chemotherapy in breast cancer patients. The reasons behind the resistance were investigated by screening epigenetically regulated genes through analysis of methylation data from tamoxifen-resistant MCF-7 cells. *MMP1* locus was found to be hypomethylated at a promoter CpG site and its expression was upregulated in the cell line, which was verified by the drug-resistant tumor tissues from breast cancer patients (*n* = 28). Downregulating MMP1 using a short hairpin RNA inhibited the growth of resistant cells and increased sensitivity to tamoxifen in vitro as well as in a xenografted mouse model in vivo. This study suggests that MMP1 is potentially a target gene to control tamoxifen resistance in breast cancer.

**Abstract:**

Background: Tamoxifen (tam) is widely used to treat estrogen-positive breast cancer. However, cancer recurrence after chemotherapy remains a major obstacle to achieve good patient prognoses. In this study, we aimed to identify genes responsible for epigenetic regulation of tam resistance in breast cancer. Methods: Methylation microarray data were analyzed to screen highly hypomethylated genes in tam resistant (tamR) breast cancer cells. Quantitative RT-PCR, Western blot analysis, and immunohistochemical staining were used to quantify expression levels of genes in cultured cells and cancer tissues. Effects of matrix metalloproteinase-1 (MMP1) expression on cancer cell growth and drug resistance were examined through colony formation assays and flow cytometry. Xenografted mice were generated to investigate the effects of MMP1 on drug resistance in vivo. Results: MMP1 was found to be hypomethylated and overexpressed in tamR MCF-7 (MCF-7/tamR) cells and in tamR breast cancer tissues. Methylation was found to be inversely associated with MMP1 expression level in breast cancer tissues, and patients with lower MMP1 expression exhibited a better prognosis for survival. Downregulating MMP1 using shRNA induced tam sensitivity in MCF-7/tamR cells along with increased apoptosis. The xenografted MCF-7/tamR cells that stably expressed short hairpin RNA (shRNA) against MMP1 exhibited retarded tumor growth compared to that in cells expressing the control shRNA, which was further suppressed by tam. Conclusions: MMP1 can be upregulated through promoter hypomethylation in tamR breast cancer, functioning as a resistance driver gene. MMP1 can be a potential target to suppress tamR to achieve better prognoses of breast cancer patients.

## 1. Introduction

Breast cancer (BC) is one of the most common cancers in women, with over two million new cases (11.7% of all cancers) and causing the death of approximately 0.7 million patients (6.9% of all cancers) worldwide in 2020 [1]. BC is caused by the dysregulation of various cancer-related genes in different cell types; therefore, it is characterized by multiple subtypes [2]. Precise characterization of the cancer type is essential to determine clinical treatment options. Presence or absence of the estrogen receptor (ER) is a crucial factor to classify the breast cancers into ER-positive (ER+) and ER-negative (ER-), with the former accounting for approximately 70% of all breast cancer cases [3]. ER+ cancer cells express a functional ER that is essential for their survival. Therefore, the ER can be considered a major target for therapeutic medicines. Tamoxifen is an antagonist of the ER, blocking its signaling activity entirely by binding to it [4]. The downstream signaling pathways including PI3K/AKT and NFκB are consequently inhibited, leading to dysregulation of cancer-related genes such as HOXA5, TNF, and CCL2 [5,6]. Eventually, proliferation of cancer cells is suppressed by induction of cell death. Based on these anticancer activities, tam has been widely used to treat ER+ breast cancer and has exhibited high effectiveness during the initial treatment period [7]. However, as observed for many other therapeutic drugs, tam-resistant (tamR) cells can emerge, which escape the drug, causing cancer recurrence [8]. It has been shown that approximately 25% of tam-treated patients experience recurrence after five years of chemotherapy [9].

Tam resistance can emerge in various ways due to dysregulation of cell proliferation-related pathways [10]. This mechanism is closely associated with CYP2D6, which is crucial for the metabolism of tam to its active metabolite [11]. Genetic variants of CYP2D6 can lead to diminished or even no enzyme activity resulting in tamR ER+ breast cancer patients [12]. The status of ERα, which is the primary target of tam, can determine a clinical outcome [13]. Typically, patients with tumors expressing low or diminished ERα levels do not benefit from tam therapy, though a small fraction of ERα-negative tumors have exhibited sensitivity to tam [14]. Tam resistance can also emerge through a ligand-independent pathway, where the ER is downregulated by receptor tyrosine kinases such as the epidermal growth factor receptor (EGFR), ErbB receptor 2 (ERBB2), and insulin-like growth factor receptor (IGF1R) [15,16]. The AKT and MAPK pathways activated by these receptors downregulate ER expression, contributing to tam resistance [17].

Epigenetic changes, including altered methylation levels at CpG sites that are usually located on the promoters of coding genes, and dysregulation of microRNAs (miRNAs), have been also implicated in several tamR cases [18,19]. Aberrant CpG island methylation has been observed at the promoter of various genes in tamR cancer cells, resulting in dysregulated transcription [20,21]. A study has demonstrated that DNA hypermethylation occurs predominantly at the estrogen-responsive enhancers and is associated with reduced ESR1 binding and decreased expression of crucial ERα activity regulators, thereby abating endocrine responses in ERα-positive breast cancers [22]. Three miRNAs, miR-101, miR-206, and miR-24-3p, have been identified to significantly increase in breast cancer tissues, with their ectopic expression inducing tam resistance in cancer cells [23,24,25]. In another study, exosomal miR-205 that targets the E2F1 transcription factor, which is a major driving force for the cell cycle, has been shown to be dysregulated in breast cancer cells by promoting tam resistance [26]. Histone modifications can also contribute to tam resistance by affecting gene expression. In tamR breast cancer cells, ERα expression has been found to be increased due to elevated H3K4 methylation levels, which are induced by the upregulation of two histone methyltransferases, MLL3 and SET1A [27].

Unlike dysregulated molecular events in tamR cancer cells, not much is known about epigenetically regulated oncogenes that can be potential chemotherapy targets. In this study, we screened genome-wide methylation data obtained from tamR MCF-7 breast cancer cells to identify hypomethylated oncogenes that were consequently upregulated, potentially aggravating the tumor state. The *matrix metalloproteinase-1 (MMP1)* gene was selected because it ranked high in the hypomethylated gene list; its epigenetic regulation was also validated in tamR breast cancer patients. MMP1 is suggested as a potential prognostic factor for the malignancy risk of cancer, as it is activated and overexpressed by many signaling pathways involved in the initiation and progression of cancer, which can promote the hallmarks of cancer such as angiogenesis, metastasis, and invasion [28]. The contribution of MMP1 in driving tam resistance was assessed by downregulating its gene expression using an shRNA. Finally, the involvement of MMP1 in tumor cell growth and drug resistance induction was examined using a xenografted mouse model.

## 2. Materials and Methods

### 2.1. Cell Culture and Lentiviral Infection

Human breast cancer MCF-7 cells were purchased from the American Type Culture Collection (Manassas, VA, USA). Tamoxifen-resistant MCF-7 (MCF-7/tamR) cells were developed as described in our previous study [29] and further used for in vitro cell assays as well as in vivo xenografts. All cells were cultured in RPMI 1640 medium (Gibco BRL, Carlsbad, CA, USA) supplemented with 2% penicillin/streptomycin (Capricorn, Ebsdorfergrund, Germany) and 10% fetal bovine serum (Capricorn) at 37 °C in 5% CO_2_. Stable MMP1 knockdown (MCF-7/shMMP1 and MCF-7/tamR/shMMP1) and control cells (MCF-7/shNC and MCF-7/tamR/shNC) were developed by seeding 5 × 10^3^ cells/well of MCF-7 or MCF-7/tamR cells on a 96-well plate. After 24 h, the cells were infected with the shRNA-harboring lentiviral particles (Origene, Rockville, MD, USA, TL311450V and TR30021V) containing 8 μg/mL polybrene (Sigma-Aldrich, St. Louis, MO, USA), followed by puromycin (1 µg/mL) selection for 10 days.

### 2.2. Study Subjects

Tamoxifen-sensitive (*n* = 33) or -resistant (*n* = 28) patient-derived breast cancer tissues used in this research were obtained according to protocols approved by the Research Ethics Board of National Cancer Center (NCC) in Korea. All tissues were acquired from patients operated on between 2012 and 2013. The clinical information of the patients has been previously described [29].

### 2.3. Quantitative Real-Time RT-PCR (qRT-PCR) and Methylation-Specific PCR (MSP)

Total RNA and DNA were isolated from patient-derived tissues and cultured cells using the ZR-Duet DNA/RNA MiniPrep Kit (Zymo research, Irvine, CA, USA). In order to detect the relative MMP1 expression levels, cDNAs were synthesized from the extracted RNAs and then amplified before measuring gene expression, as described previously [30]. In total, 100 ng of genomic DNA was bisulfite modified using the Zymo Research EZ DNA Methylation Kit (Zymo Research, Irvine, CA, USA), and MSP was conducted as described previously [31]. Oligonucleotide primers were synthesized by IDT (Coralville, IA, USA) and Bioneer (Seoul, Korea) (Appendix A).

### 2.4. Proliferation and Tamoxifen Sensitivity Assays

The CCK-8 assay (Dojindo Laboratories, Kumamoto, Japan) and the colony formation assay were both employed to monitor cell viability and sensitivity recovery. In order to measure cell growth rate, 3 × 10^3^ cells/well were seeded in a 96-well plate and cultured for up to 6 days, before being stained with WST-8 from the CCK-8 solution. The optical density (OD) was measured at 450 nm using a microplate reader (Sunrise, Tecan, Switzerland). The tam sensitivity was evaluated by treating 1 × 10^4^ cells with tam for 24 h in 96-well plates, before adding the CCK-8 solution. After incubation for 90 min, the OD_450_ values were measured using the plate reader.

For the colony formation assay, 3 × 10^3^ cells were seeded in a 60-mm dish, treated with tam for 24 h, and then cultured in refreshed media containing no tam. Staining and colony measurements were conducted as described previously [32].

### 2.5. Western Blot Analysis

The total protein was extracted from cell lysates and Western blot analysis was performed as described previously [32]. Briefly, the breast cancer cells were suspended in RIPA buffer containing 1% Halt protease inhibitor cocktail (Thermo Fisher Scientific, Waltham, MA, USA). The protein (35 μg) was blotted with either anti-MMP1 (1:400, Bioss, Woburn, MA, USA, bs-0424R) or anti-β-actin (1:1000, Bioss, bs-0061R) overnight, followed by incubation with an HRP-conjugated anti-rabbit IgG antibody (1:1000, Genetex, Irvine, CA, USA, GTX213110-01) for 2 h. The bands were visualized using the ECL reagent (Abfrontier, Seoul, Korea).

### 2.6. Apoptosis Assay

For detection of apoptosis and necrosis, the APC Annexin V Apoptosis Detection Kit with propidium iodide (PI) (BioLegend, San Diego, CA, USA) was used, following the manufacturer’s instructions. The harvested cells were washed twice with PBS and resuspended in binding buffer at a density of 1 × 10^6^ cells/mL. The cells were treated with Annexin V and the PI reagent and incubated for 15 min in the dark. Samples were assayed using an Accuri C6 flow cytometer (BD Biosciences, San Jose, CA, USA).

### 2.7. Xenograft Mouse Model

All animal studies were approved by the Institutional Animal Care and Use Committee of Dongguk University (No: IACUC-2017-010-1). In total, 1 × 10^7^ cells (MCF-7/tamR/shNC and MCF-7/tamR/shMMP1) in a 100-μL suspension [1:1 mix of PBS and Matrigel (BD Biosciences, San Jose, CA, USA)] were implanted subcutaneously into 6-week-old female BALB/c nude mice (Orient Bio, Sungnam, Korea), followed by implantation of 60-day release 17β-estradiol pellets (0.72 mg/pellet total dose; Innovative Research of America, Sonnasota, FL, USA). After 4 weeks, the test group was intraperitoneally injected with 100 μL (1 mg/kg) of tam in corn oil (Sigma-Aldrich, St. Louis, MO, USA) five times per week for 4 weeks, whereas the control group was treated with only corn oil. Tumor volume (length × width^2^ × 0.5) was measured each week. After eight weeks of cell injection, xenografted tumors were collected. The tissues were fixed in 4% paraformaldehyde and embedded in paraffin blocks for histological analysis.

### 2.8. Immunohistochemistry (IHC) Staining

IHC staining was conducted to examine MMP1 expression in mice xenografts. After deparaffinizing tissue sections, endogenous peroxidase blocking was conducted for 10 min, followed by proteinase blocking for 1 h before incubation with the rabbit anti-MMP1 antibody (1:200, Bioss, bs-0424R). The EnVision Detection System (K5007, Agilent Dako, Santa Clara, CA, USA) was used for detection. Sections were counterstained using hematoxylin, hydrated, cleaned, and mounted under coverslips. Slides were visualized using an Olympus BX41 light microscope. The 3,3′-diaminobenzidine (DAB)-positive areas were determined from IHC images using the ImageJ program (v.1.51j8) (https://imagej.nih.gov/ij/download.html (accessed on 6 September 2021)).

### 2.9. Data Mining and Statistical Analysis

Genes showing a significant hypomethylation in the MCF-7/tamR cells were retrieved from the methylation array data of the NCBI GEO DataSet (GSE132615 and GSE132616). The association of MMP1 expression level with the clinical breast cancer patient outcome was investigated using the GOBO tool (http://co.bmc.lu.se/gobo (accessed on 4 November 2021)). The MethHC database (http://methhc.mbc.nctu.edu.tw (accessed on 4 November 2021)) was used to analyze the correlation between promoter methylation and MMP1 expression level. Data from qRT-PCR, MSP, Western blotting, and IHC analysis and those from flow cytometry and xenograft were statistically analyzed using the Student’s *t*-test and two-way ANOVA followed by the Tukey post hoc test, respectively (significance threshold: *p* < 0.05); presented values represent the mean ± SEM. Statistical analysis was performed using SPSS software for Windows, release 17.0 (SPSS, Chicago, IL, USA).

## 3. Results

### 3.1. MMP1 Is Upregulated by DNA Hypomethylation in tamR Breast Cancer

In our previous study, genome-wide methylation was analyzed in MCF-7/tamR cells to identify epigenetically regulated genes responsible for drug resistance [29]. In the current study, we focused on hypomethylated genes, as numerous oncogenes have been reported to be upregulated by hypomethylation. The 20 most hypomethylated genes were screened from the array data by comparing the CpG methylation levels between the MCF-7/tamR cells and the parental tam-sensitive MCF-7 cells (Figure 1A). The alteration in methylation level (Δβ) of the 20 genes ranged from −0.47 to −0.22 (Table 1). MMP1 with a Δβ of −0.32 was selected for further investigation because it is an oncogene known to be crucial in tumor development and metastasis [33]; however, its epigenetic regulation and molecular mechanisms that contribute to chemo-resistance have not been elucidated.

In order to confirm hypomethylation at the specific CpG site of MMP1, its methylation level was examined from the chromosomal DNA of MCF-7/tamR cells by performing methylation-specific PCR. The results revealed a methylation decrease of 73% in tamR cells compared to that in parental MCF-7 cells (Figure 1B). MMP1 gene and protein expressions were determined using qRT-PCR and Western blot analysis, which revealed upregulation of the MMP1 transcript as well as the MMP1 protein in tamR cells (Figure 1C,D; Appendix A). Subsequently, the methylation and expression levels were examined in tumor tissues obtained from breast cancer patients who underwent surgery at the National Cancer Center, Korea. The tamR tissues (*n* = 28) were found to exhibit lower methylation (*p* < 0.005) and higher MMP1 expression than those in the tam-sensitive (tamS) tissues (*n* = 33, *p* < 0.005) (Figure 1E). The methylation and MMP1 expression levels were found to be negatively correlated in the breast cancer tissues (*n* = 61; *r* = −0.35; *p* < 0.05) (Figure 1F). MMP1 expression levels in breast cancer patients also affected patient prognosis for survival, with shorter survival periods in patients with high MMP1 expression than those in patients with low MMP1 expression (Figure 1G). These data indicate that MMP1 expression is increased by promoter hypomethylation in tamR breast cancer patients compared to that in tamS cancer patients, potentially affecting patient prognosis and chemotherapeutic options.

### 3.2. MMP1 Stimulates MCF-7/tamR Cell Proliferation and Enhances Tam Resistance

MMP1 is an oncogene involved in various cancers including breast cancer [34]; however, its role in chemotherapeutic drug resistance remains undetermined. Therefore, its effects on cancer cell growth and drug resistance were examined in tamR and tamS cancer cells. First, cell growth and drug resistance were examined in MCF-7/tamR cells, wherein MMP-1 was downregulated by stably transfecting an MMP1 shRNA (shMMP1). (Appendix A). The control MCF-7/tamR/shNC cells were stably transfected with a control shRNA (shNC). The shMMP1-transfected cells exhibited retarded growth compared to that of the control shRNA-transfected cells (Figure 2A). Furthermore, the shMMP1-transfected cells displayed more sensitivity to tam (<1.0 μM) than that displayed by control cells (Figure 2B). These results were also observed in the parental MCF-7 cells that have lower MMP1 expression than that of tamR cells; however, the change in expression level was not significant (Figure 2C,D). Then, the effects of MMP1 on drug resistance were assessed through a colony formation assay. Downregulating MMP1 markedly inhibited colony growth even in the absence of tam, which muddled the results of the effects of tam on drug resistance (Figure 2E). Inhibiting MMP1 expression in MCF-7 cells resulted in the clearly observed recovery of tam sensitivity (Figure 2F).

In order to examine the antigrowth effect of MMP1 downregulation combined with tam treatment on apoptosis or necrosis, the MCF-7/tamR cells harboring either a control shRNA (MCF-7/tamR/shNC) or the shMMP1 (MCF-7/tamR/shMMP1) were examined. Apoptosis and necrosis of cells were found to increase by 8.2 and 0.6%, respectively, in the MCF-7/tamR/shMMP1 cells compared to those in the control cells. Apoptosis and necrosis further increased by 7.5 and 1.5%, respectively, upon tam treatment, while the number of live cells decreased (Figure 3). Taken together, these results indicate that MMP1 is a crucial oncogene to stimulate MCF-7 cancer cell growth and a driving factor for drug resistance.

### 3.3. MMP1 Increases Tumor Growth of the MCF-7/tamR Cells in Xenografted Mice

The aforementioned in vitro observations were tested in vivo by evaluating whether MMP1 can drive tumor cell growth and drug resistance in a xenograft mouse model. First, MMP1 expression was examined in the tumor tissues obtained after xenografting MCF-7 and MCF-7/tamR cells into nude mice, as described in our previous study [29]. In situ immunohistochemistry of tumor sections validated the elevated MMP1 expression in MCF-7/tamR cells compared to that in parental MCF-7 cells (Figure 4A,B).

Subsequently, the involvement of MMP1 in tumor growth and tam resistance was investigated in the xenografted mice. Both MCF-7/tamR/shMMP1 tumor cells and MCF-7/tamR/shNC cells were transplanted intraperitoneally into mice, and the tumor growth was monitored for 8 weeks. The MCF-7/tamR/shMMP1 cells expressing low MMP1 levels exhibited dramatically retarded tumor growth by 79% at week 8 compared to the control cells (*n* = 5, *p* < 0.05) (Figure 4C,D). Other groups were intraperitoneally administered with tam after 4 weeks of transplantation to examine the effects of MMP1 on drug sensitivity recovery. Tumor sizes in tam-treated mice were significantly smaller (>7-fold) compared to that of untreated control group (*n* = 5, *p* < 0.05) (Figure 4E,F). The MCF-7/tamR/shMMP1 cells without tam treatment were found to grow slower than those treated with tam; however, the growth was reversed after 8 weeks (Figure 4G).

## 4. Discussion

In this study, we aimed to identify signature genes that contributed to the emergence of tam resistance through epigenetic regulation in breast cancer. MMP1 was identified after a thorough examination through a series of experiments. The *MMP1* gene was found to be markedly hypomethylated at a specific CpG site of its promoter, while its transcript and protein levels were found to be elevated in MCF-7/tamR cells. It is known that tam is metabolized to hydroxytamoxifen (OH-tam) in vivo, which has 20~25 times higher affinity to ER, although both forms bind specifically to ER [35]. Therefore, use of OH-tam instead of tam in the cultured MCF-7 cells in vitro could result in stronger cellular as well as molecular effects. A negative association between promoter methylation and MMP1 expression levels was also confirmed in tamR and tamS breast cancer patients. Downregulation of MMP1 using shRNA inhibited the growth of the MCF-7/tamR cells in vitro as well as that of the tumor tissues derived from the MCF-7/tamR cells in the animal model. Although the orthotopic injection reproduces a more realistic environment of the breast cancer, the subcutaneous approach adopted in this study was enough to explain the efficacy of MMP1 in stimulating the tumor cell growth and drug resistance. Furthermore, breast cancer patients with high MMP1 expression exhibited a lower survival ratio compared to those with low expression. Collectively, these experimental results indicated that oncogenic MMP1 causes cancer cells to acquire chemo-resistance during chemotherapy.

MMP1 is a member of the MMP family of zinc-containing endopeptidases that have multiple functions in tissue remodeling, including their participation in turnover of collagen fibrils in extracellular spaces and the cleavage of nonmatrix substrates [36]. The protein has been shown to be upregulated in various cancers including breast cancer, and its expression has a significantly negative correlation with patient survival [37]. Previous studies have suggested that *MMP1* can be epigenetically regulated; a methyltransferase inhibitor, 5-aza-2′-deoxycytine, has been shown to induce a decrease in its protein expression in a human fibrosarcoma cell line [38]. The *MMP1* gene promoter does not appear to contain CpG islands [38], suggesting the presence of a single or a few CpGs at specific loci for regulation of gene expression. Here, the methylation array data and bisulfite-PCR assay revealed a single CpG at the promoter, highlighting a significant correlation between its methylation and MMP1 expression levels in tamR and tamS breast cancer cells and tumor tissues.

Although the molecular mechanism of tam resistance has implicated multiple signaling pathways, the mechanism by which MMP1 induces tam resistance remains elusive. The activity of MMP1 in tam resistance can be speculated through two mechanisms. First, MMP1 can crosstalk with the ER+ pathway. Several previous studies have demonstrated the regulation of MMP1 by ER. For instance, Thaler et al. reported increased expression of transfected MMP1-promoter constructs in response to ERβ [39]. Jung et al. revealed crosstalk between ER and human epidermal growth factor receptor 2 (HER2) to induce MMP1 expression [40]. MMP1 has also been shown to crosstalk with the IGF pathway by degrading IGFBP-1 that binds IGF with high affinity. Elevated IGFBP-1 expression has been reported in tamR MCF-7 and T-47D breast cancer cells, suggesting an association between tam resistance and IGFBP-1 accumulation [41]. Second, MMP1 might induce tam resistance by acting independently through previously identified routes including ER, EGF, or IGF pathways. This mechanism of action could be supported by the fact that most MMP1-regulating genes do not overlap with those involved in previously known pathways. For example, MMP1 is regulated by various cytokines such as epidermal growth factor, hepatocyte growth factor, and a few types of interferons [42,43,44]; however, these factors are not observed in cases with previously established tam resistance.

Elucidating downstream molecular events initiated by the upregulation of MMP1 should be the next step to better understand the mechanism by which the gene induces tam resistance in breast cancer. Recently, Hamadneh et al. reported dysregulation of PI3K/AKT and MAPK1 in the tamR-MCF-7 cells [45]. These genes are known to be regulated by MMP1 and their increased expression has been shown to promote proliferation and metastasis of cancer cells [46]. Notably, hypomethylation was observed at numerous CpG sites including the MMP locus during investigation of tam resistance. SRPX2, LNX1, GUCY1B3, and CD59 are among the top 20 hypomethylated genes and are known to be oncogenes. CD59 has been shown to be elevated in tamR MCF-7 cells, validating our methylation array data. SRPX2 has also been shown to confer chemoresistance for 5-Fu and gemcitabine by activating the PI3K/AKT axis in pancreatic cancer [47]. LNX1 has been shown to contribute to cisplatin resistance by regulating cell cycle progression in human cancer cells [48]. For all these genes, epigenetic modulation of gene expression involved in chemoresistance has not been elucidated. Our methylation data suggest that many oncogenes are upregulated due to hypomethylation in chemo-resistant cancers including tamR breast cancer. Moreover, the PI3K/AKT pathway functions as a linchpin for the emergence of various types of chemoresistance. A limitation of this study is the lack of experimental evaluation of the impact of *MMP1* methylation status on gene expression and tam resistance. Nonetheless, our results revealed a strong association between the methylation level and expression, especially in the tamR and tamS breast cancer patients. Furthermore, we demonstrated that MMP1 is a driving factor to induce tam resistance using transplanted cancer cells.

## 5. Conclusions

MMP1 is a gene that drives breast cancer cells to become tamR. During the induction of tam resistance, the *MMP1* gene is upregulated through hypomethylation at its promoter. Suppressing MMP1 expression attenuates tam resistance and increases apoptosis, as confirmed by in vitro cell culture and an in vivo xenograft mouse model. Taken together, these results confirm that MMP1 is a crucial gene involved in induction of tam resistance; therefore, it could be used as a molecular target to prevent or treat tamR breast cancer.

## Figures and Tables

**Figure 1 cancers-14-01232-f001:**
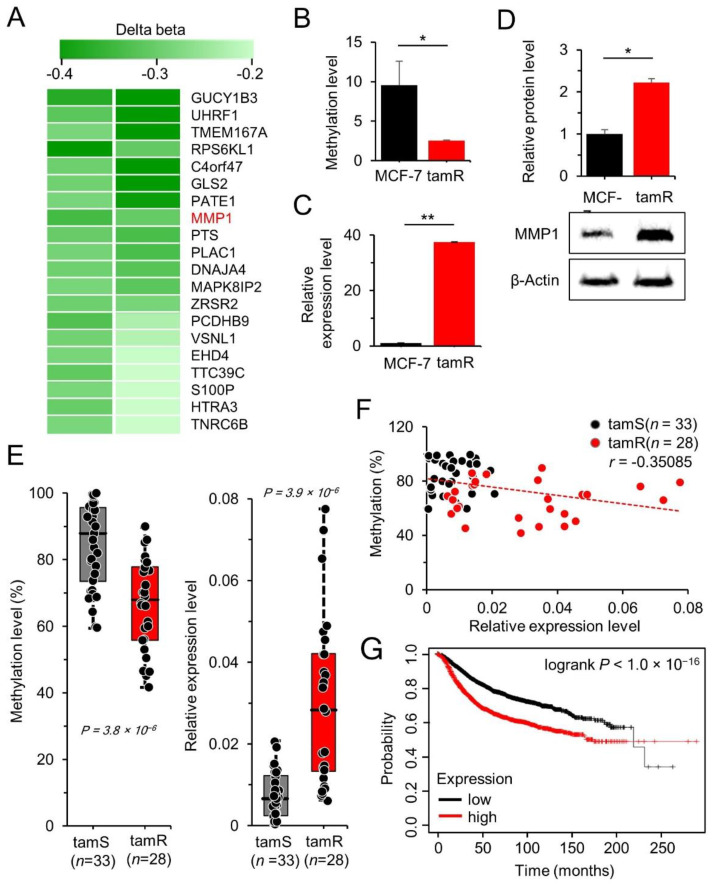
*MMP1* is hypomethylated and upregulated in tamR breast cancer compared to that in tamS breast cancer. (**A**) Heatmap of the 20 most hypomethylated genes in MCF-7/tamR cells compared to those in parental MCF-7 cells. Data of two arrays are presented. Methylation level of the CpG site at the MMP1 promoter (**B**) and mRNA level (**C**) were analyzed by MSP and qRT-PCR in breast cancer cells. The data are presented as mean ± SE of three independent experiments. (**D**) Western blot analysis was performed to detect MMP1 protein level in cultured cells. (**E**) Hypomethylation and upregulation of MMP1 were identified by MSP and qPCR in breast cancer tissue of patients. *n:* number of samples. (**F**) The association between the CpG methylation and mRNA expression of MMP1. (**G**) Kaplan–Meier survival curve of MMP1 level in breast cancer patients. Samples (*n* = 1379) were categorized into tertiles based on MMP1 expression. Distant metastasis-free survival (DMFS) was compared for all tumor samples using the log-rank test (*p* < 0.001). * *p* < 0.05, ** *p* < 0.01.

**Figure 2 cancers-14-01232-f002:**
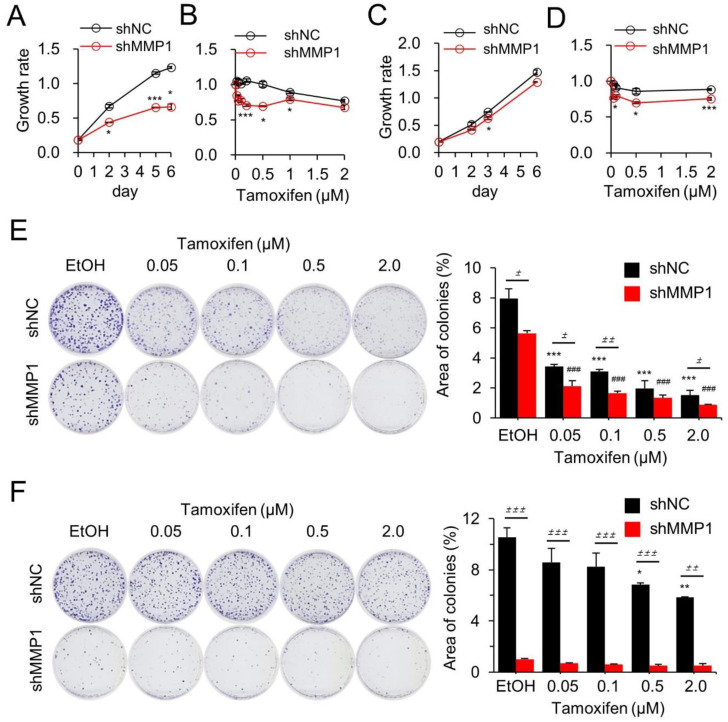
MMP1 promotes cell growth and confers tam resistance in MCF-7 and MCF-7/tamR cells. MMP1 was downregulated by injecting a stably expressed shRNA lentiviral vector into MCF-7 and MCF-7/tamR cells. Effects of MMP1 on cell growth (**A**,**C**) and tam sensitivity (**B**,**D**) were examined by colorimetric analysis using the CCK-8 reagent in MCF-7/tamR (**A**,**B**) and MCF-7 cells (**C**,**D**). Effects of MMP1 on tam sensitivity were confirmed by the colony formation assay for MCF-7 (**E**) and MCF-7/tamR cells (**F**). All assays were performed in three independent experiments. Data are presented as mean ± SE. The values at each day of culture were compared to assess statistical significance in (**A**−**D**). Representative images are shown for the colony formation assay. shMMP1, short hairpin RNA against MMP1; shNC, negative control shRNA. * *p* < 0.05, ** *p* < 0.01, and *** *p* < 0.001 versus shNC control; ^###^
*p* < 0.005 versus shMMP1 control; ^±^
*p* < 0.05, ^±±^
*p* < 0.01, ^±±±^
*p* < 0.005.

**Figure 3 cancers-14-01232-f003:**
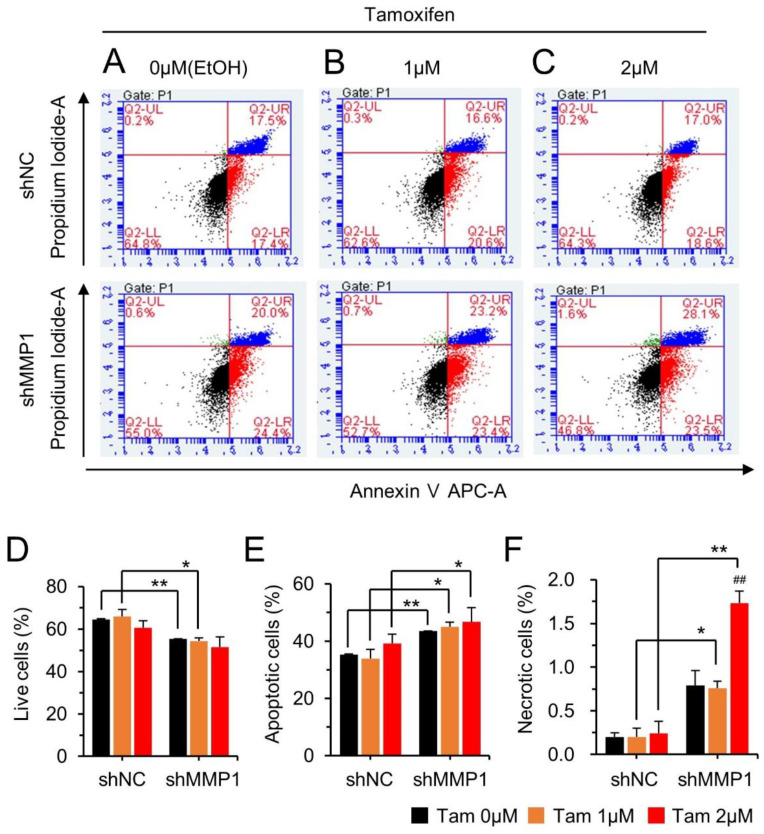
Effects of MMP1 on apoptosis and necrosis of MCF-7/tamR cells. Flow cytometry was performed after downregulating MMP1 with a stably expressed shRNA lentiviral vector. Representative flow cytometry images at 0 μM (**A**), 1 μM (**B**), and 2 μM (**C**) of tam are shown. Top and bottom images are for shNC and shMMP1, respectively. Cells in each quadrate of the FACS image represent, clockwise from the upper left, necrosis (green), late apoptosis (blue), early apoptosis (red), and live cells (black). Ratio of live cells (**D**), apoptotic cells (**E**), and necrotic cells (**F**) are shown in a bar graph. All assays were performed for three independent experiments, and data presented as mean ± SE. shMMP1, short hairpin RNA against MMP1; shNC, negative control shRNA. * *p* < 0.05, ** *p* < 0.01; ^##^
*p* < 0.01 versus shMMP1 control.

**Figure 4 cancers-14-01232-f004:**
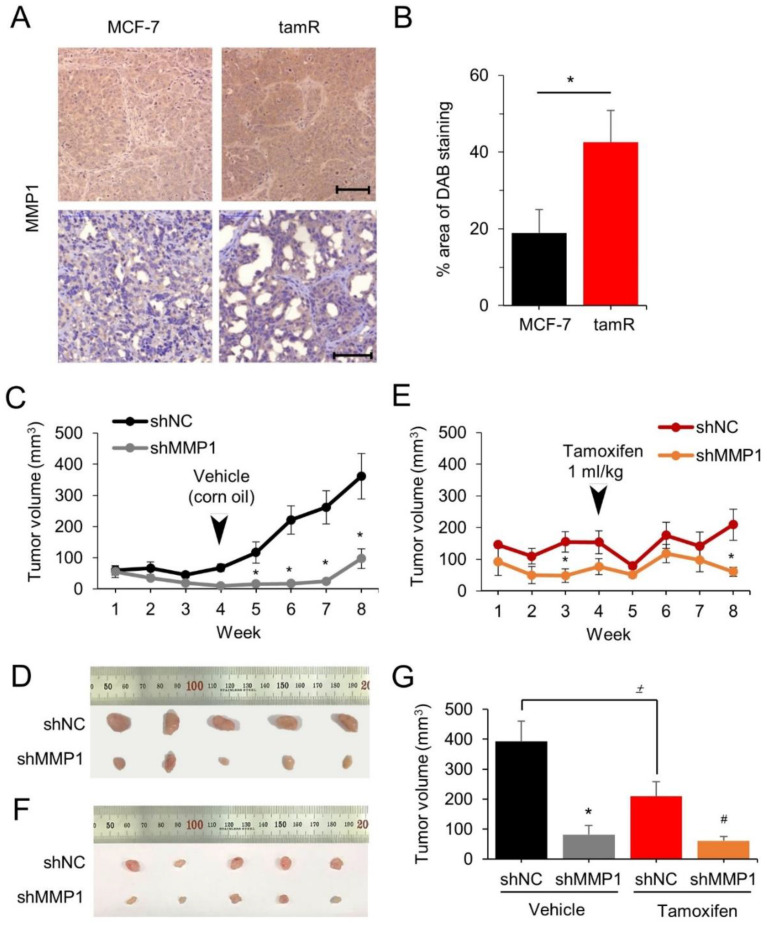
Downregulating MMP1 inhibits tumor growth and induces tam sensitivity in xenografted mice. (**A**) Immunohistochemical analysis of MMP1 expression was conducted on xenografted tumor tissues containing MCF-7 and MCF-7/tamR cells. Three tissue sets were analyzed, and the protein expression is depicted in the bar graph (**B**). Representative images are shown. Scale bar, 100 μm. MCF-7/tamR cells stably transfected with shMMP1 or shNC were subcutaneously injected into BALB/c nude mice. The tumor volume was examined every week for eight weeks. Corn oil (**C**) or Tam (**E**) was administered according to intraperitoneal method after 3 weeks of cell transplantation. At week 8, mice were sacrificed to extract untreated (**D**) and tam-treated tumor tissues (**F**), of which size is denoted in a bar graph (**G**). *n* = 5 for each group. * *p* < 0.05 versus shNC control; ^#^
*p* < 0.05 versus shNC tam; ^±^
*p* < 0.05.

**Table 1 cancers-14-01232-t001:** Top twenty hypomethylated genes in the MCF-7/tamR cells.

Gene Symbol	Accession No.	Description	Δ β-Value ^a^	Fold Change ^b^
*GUCY1B3*	NM_000857	guanylate cyclase 1 soluble subunit beta 1	−0.47	−2.24
*UHRF1*	NM_001048201	ubiquitin-like with PHD and ring finger domains 1	−0.42	−2.37
*TMEM167A*	NM_174909	transmembrane protein 167A	−0.37	−2.32
*RPS6KL1*	NM_031464	ribosomal protein S6 kinase-like 1	−0.36	−2.64
*C4orf47*	NM_001114357	chromosome 4 open reading frame 47	−0.35	−1.87
*GLS2*	NM_013267	glutaminase 2	−0.34	−2.15
*PATE1*	NM_138294	prostate and testis expressed 1	−0.33	−1.60
*MMP1*	NM_002421	matrix metallopeptidase 1	−0.32	−1.96
*PTS*	NM_000317	6-pyruvoyltetrahydropterin synthase	−0.31	−3.70
*PLAC1*	NM_021796	placenta enriched 1	−0.30	−2.10
*DNAJA4*	NM_018602	DnaJ heat shock protein family (Hsp40) member A4	−0.30	−1.62
*MAPK8IP2*	NM_012324	mitogen-activated protein kinase 8 interacting protein 2	−0.29	−1.87
*ZRSR2*	NM_005089	zinc finger CCCH-type, RNA binding motif and serine/arginine rich 2	−0.29	−1.54
*PCDHB9*	NM_019119	protocadherin beta 9	−0.27	−3.16
*VSNL1*	NM_003385	visinin-like 1	−0.25	−1.66
*EHD4*	NM_139265	EH domain containing 4	−0.24	−1.88
*TTC39C*	NM_153211	tetratricopeptide repeat domain 39C	−0.24	−1.94
*S100P*	NM_005980	S100 calcium binding protein P	−0.23	−1.54
*HTRA3*	NM_053044	HtrA serine peptidase 3	−0.22	−1.61
*TNRC6B*	NM_001024843	trinucleotide repeat containing adaptor 6B	−0.22	−2.24

^a^ The values were obtained by subtracting the average methylation level of MCF-7/tamR cells with that of MCF-7 cells from two independent array datasets (GSE132615 and GSE132616). ^b^ The values were obtained by dividing the average methylation level of MCF-7 cells by that of MCF-7/tamR cells from two independent array datasets (GSE132615 and GSE132616). A negative symbol is added to the obtained value.

## Data Availability

Publicly available datasets were analyzed in this study. These data can be found here (https://www.ncbi.nlm.nih.gov/geo/query/acc.cgi?acc=GSE132615/GSE132615 (accessed on 12 July 2021) and GSE132616).

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
