# Peer review of "Matrix Metalloproteinase-1 (MMP1) Upregulation through Promoter Hypomethylation Enhances Tamoxifen Resistance in Breast Cancer"

_cancers, 2022, doi:10.3390/cancers14051232_

Round 1

Reviewer 1 Report

The authors desribed the role of MMP1 overexpression in the development of resistance of breast cancer cells to tamoxifen and suggested that this protein might become a therapeutic target. The study is well-designed and is based on the appropriate methodology. Outcomes of the work have potential clinical consequences. Manuscript is well-written in an understandeable manner with good language quality. I have several comments, mainly focused on the improvement of the analysis and interpretation of obtained data:

1) Revise the text and remove spelling mistakes (such as sort instead of short in lines 264 and 280).

2) Full names for OD (optical density?; line 136) and DAB (line 180) must be introduced.

3) Table 1, lines 207 and 209: Could authors explain, why only 2 independent array data were used? Normally, 3 independent repetitions are accepted as minimum for proper data analysis and raising of correct conclusions.

4) Line 213: The formulation is not correct. Methylation level decreased from approx. 9 to 2, i.e. not by 7 %, but by 78 %.

5) Fig. 1D: Instead of percentage (121%), significancy star(s) should be above the line.

6) Line 233: Authors must indicate, which type of replicate they mean. I assume, that they mean independent repetitions (otherwise, if they mean just biological replicates, their data would not be sufficiently robust). This mistake is throughout the manuscript, please revise this issue comprehensively.

7) Fig. 2 A, B, C, D: SE bars are missing, please add them. Statistics is missing, without it, you are not allowed to derive the conclusions presented in the respective Results text. Please, add statistical analysis.

8) Fig. 2E, F right side: The same problem as point 6.

9) Line 270, „….in the control cells (MCF-7/tamR).“: I am not sure, but should not be MCF-7/tamR replaced with MCF-7/tamR/shNC? Not presenting shNC can lead to confusion that you mean non-transduced resistant MCF-7s.

10) Figure 3 and related text need significant revision. First, statistical analysis is missing in D, E, F (see my point 6), add it and revise Results text. The authors present comparisons with shNC-transduced cells in Results section, but the 1) data from them are missing in figure (introduce them) and 2) statistical analysis must be present for this comparison. Generally, I have a problem with the impact of data as in D, E, I do not see any significant changes on a first view. Possibly significant changes are in F, but overal necrotic percentage is so low, that it is questionable, whether it has some consequence. In addition, necrosis is unspecific death type, while rather appoptosis is related to cancer growth/death (in apoptosis graph, small changes are visible, significancy is questionable until not done). Lastly, for less experienced readers, description of what you measure in which quadrate of A, B, C (i.e. upper left apoptosis, etc.) should be mentioned in figure legend.

11) Fig. 4B is missing statistics. You are not allowed to talk about significant changes in Results section until you perform the analysis. Furthermore, in the Results text, authors try to compare variants with and without tamoxifen treatment. They are not allowed as data are separated into two subgraphs (C and E) and statistical analysis is missing. I understand that it would not be nicely readable if they fuse C and E, but at least new graph showing data exclusively from end-point, i.e. 8th week, should be generated next to them and data should be statistically analysed in the new graph.

12) Line 302: Authors present 66%, but if I look on figure, I see values around 360 mm3 vs. 100 mm3, i.e. change is around 72%.

13) Lines 307 – 310: This sentence is strange as anyway, authors discussed these data elsewhere in the text in different comparison. There is no inrobust data in 8th week, which would not be available for analysis/discussion.

Reviewer 2 Report

The present manuscript describes the study of the importance of matrix metalloproteinase-1 (MMP1) upregulation in tamoxifen resistant breast cancer cell line MCF-7 and the respective implications of the downregulation of the gene using a short hairpin RNA.

The summary and abstract are well written and clearly translate the goals of this study and methodologies used that led to the final conclusions. Nevertheless, the introduction section lacks of a general description on breast cancer incidence and mortality. These numbers would help the audience to understand the medical need underlying the current treatments used in clinics and to identify the number of patients that would actually benefit from a new breakthrough in cancer resistance. In general, this section needs more literature support, there are many reports and reviews concerning MMP1 role in cancer.

In the first part of the results section, the authors establish a correlation between in vitro results and patients derived tissue, leading to the conclusion of a higher expression in MMP1 and lower methylation status. The study using shMMP1 cells is presented , but appears incomplete from my point of view. Figures 2 (E and F), 3 (D, E and F) and 4 (B) describe several assays but no statistic treatment was made of the results. If the presented assays are representative of 3 independent experiment (as it should) in the presence of respective control groups, full statistical analysis is crucial to validate the presented conclusions.

From the second in vivo study, authors report that due to the size of tumors, no clear conclusion could be made. Why was this experience stopped at this time and not prolonged until tumor sizes would have allowed a conclusion on this matter?

Discussion is also lacking more literature support and in vivo results were not included.

This study would benefit in overall impact for the scientific community if the authors would add statistical validation over all the findings and some more support from literature.

Reviewer 3 Report

This paper seeks to find an explanation for tamoxifen resistance of breast cancer cells. They concentration on the role of hypomethylation at a promoter CpG site of MMP-1 and its expression was upregulated in the cell line. This observation was supported by the drug-resistant tumor tissues from breast cancer patients. Downregulating MMP1 using a short hairpin RNA inhibited the growth of resistant cells and also increased sensitivity to tamoxifen in vitro as well as in a xenografted mouse model. The authors suggest that MMP1 is potentially a target gene to control tamoxifen resistance in the breast cancer.

  • The in vitro experiments use tamoxifen rather than its active metabolite, 5-hydroxytamoxifen. How do we know that several of the effects do not depend on the ability to convert tamoxifen to 5-hydroxytamoxifen?
  • The authors ought to acknowledge that the repeated i.p. injection of corn oil causes an inflammatory response that could affect their results. Molecular Cancer Researchdoi: 1158/1541-7786.MCR-20-0650
  • To establish that the MMP-1 antibody is suitable for IHC, the authors need to show a complete Western blot of extracted samples in Supplementary results to establish mono-specificity.
  • Statistics: The only analysis that is described is a Student’s t test but this cannot be used for multiple comparisons which are made. The authors need to show appropriate statistical analysis of all results in the Figures. Each legend should describe the numbers of independent experiments, which defines the n value. Replicates give the mean result for each experiment and cannot be added to increase the n value.
  • 4. Evidence from Western blotting is needed to define the extent of the knockdown of MMP-1. Were other shRNA used because it is normal to show knockdown with different constructs.
  • 3. Please, justify how apoptosis and necrosis identified and differentiated separately?
  • It seems to be a pity that the authors did not use an orthotopic breast cancer model because flank injections of cancer cells do not reproduce the real environment for the breast cancer.
  • 4. Three sections were analyzed, whereas 5 mice are the minimum number that are recommended for use in animal experiments. How many sections from each tumor analyzed and how was the section chosen? Fig. 4 D and F need to be quantified.

Minor comment

Line 28 and 32 “were” not “was”.

Round 2

Reviewer 1 Report

The authors properly addressed majority of my comments. I have only minor remaining criticism to be considered:

1) To my previous point 6: Authors specified type of replicate only in Fig. 1, but did not follow my commentary in other cases, as I requested. Please add word „independent“ preceding word „replicate“ also in legend to Fig. 2 and 3 to make this issue clear to readers.

2) Figure 2: You should specify, what you compare with what within statistical analysis. Either do it by showing connecting lines in the figure (which is not convenient in A, B, C, D, but suitable for E, F) or describe it by words in the legend (use this variant for A, B, C, D). I assume it is particular shNC vs shMMP1C variants within individual days (in Fig. 2A, B, C, D) and shNC vs shMMP1 within individual tamoxifen concentrations (Fig. 2 E, F – here it would be easier to do connecting lines instead of description). As a reader, I should not assume/estimate, but be sure what’s compared based on your clear desription.

3) To my previous comment 13: You show comparisons in Fig. 4G, so the last sentence in 3.3 does not make sense, as I explained in the last revision. There are no inrobust/invalid data. Please remove last sentence (lines 320 – 322 in revised version).

Reviewer 2 Report

After analysis of the correction made by the authors I recommend publication in Cancers.

The authors addressed all my comments, which improved the manuscript.

Author Response

The authors thank to the reviewer for kindly reviewing their manuscript.

Reviewer 3 Report

The authors have answered most of the criticisms from Reviewer 3 satisfactorily with the exception of Items 1 and 4.

Item 1. It is not questioned that both Tam and 4-HT both bind specifically to the estrogen receptor or that other studies have used tamoxifen itself. The point was that the binding affinity for 4-HT has been reported to be 25-50 times higher affinity than tamoxifen. Therefore, the question was whether several of the effects that are reported could not depend on the ability to convert tamoxifen to 5-hydroxytamoxifen, which is considerably more poyent? The authors have failed to address this question.

Item 4) Statistics: The only analysis that is described is a Student’s t test. Even if the studies focused on the comparison between the two groups of control vs. tamoxifen-resistant cells, or control vs. MMP1-downregulated cells etc, there are multiple comparisons in Figures 2, 3 and 4. A Student t test cannot be used for multiple comparisons. The authors need to respond to this criticism and show statistical analysis using an appropriate ANOVA with a suitable post hoc test.
